# Hierarchic superradiant phases in anisotropic Dicke model

D. K. He and Z. Song[*]

*School of Physics, Nankai University, Tianjin 300071, China*

We revisit the phase diagram of an anisotropic Dicke model by revealing the non-analyticity induced by underlying exceptional points (EPs). We find that the conventional superradiant phase can be further separated into three regions, in which the systems are characterized by different effective Hamiltonians, including the harmonic oscillator, the inverted harmonic oscillator, and their respective counterparts. We employ the Loschmidt echo to characterize different quantum phases by analyzing the quench dynamics of a trivial initial state. Numerical simulations for finite systems confirm our predictions about the existence of hierarchic superradiant phases.

## I. INTRODUCTION

With the gradual development of experiments on light-matter interaction [1–4], the quantum simulation of the Dicke model [5–8] is transitioning from theory to experiment. The Dicke model [9–13] is a fundamental model in the field of quantum optics, describing the interaction between a single-mode light field and $N$ two-level atoms. The Dicke model has a broad prospect and great potential in the field of quantum batteries [14–20]. In the thermodynamic limit ($N \to \infty$), the ground state of the Dicke model undergoes a quantum phase transition (QPT) from the normal phase (NP) to the superradiant phase (SP) [11–13, 21–23] at a certain critical coupling strength, which is referred to as the superradiant phase transition. In addition to the QPT of the ground state demonstrated above, the Dicke model also exhibits three distinct phase transitions, namely, the dissipative phase transition (non-equilibrium quantum phase transition) [24–26], the excited-state quantum phase transition [27–29], and the thermal phase transition [30, 31].

The concept of EPs [32–34], which represents the degeneracies of non-Hermitian operators, is regarded as a unique feature of non-Hermitian systems. However, subsequent research has shown that EPs exist not only in non-Hermitian systems but also in Hermitian systems [35–45]. The non-analyticity induced by EPs suggests the presence of a phase transition at this point. In previous studies, we demonstrated that the superradiant quantum phase transition in the Dicke model can be seen as the effect of two hidden second-order EPs [45, 46]. This drives us to seek a more general Dicke model to investigate its phase transitions. A more general version of the Dicke model is called the anisotropic Dicke model [20, 47–52] (ADM), in which the strengths of the rotating-wave and counter-rotating-wave terms are different. The ADM is being widely studied, including its applications in quantum batteries [20] and the ergodic-to-nonergodic transition [47, 48], as well as work related to quantum chaos [50].

In this work, we focus on the ADM Hamiltonian and identify the hidden EPs of this Hamiltonian in the ther-

* songtc@nankai.edu.cn

modynamic limit. The EPs divide the parameter space into four regions. The results show that, in addition to the existing NP to SP transition, there exists a hierarchical structure within the SP phase. In each region, the original Hamiltonian consists of different combinations of equivalent Hamiltonians, including the harmonic oscillator and the inverted harmonic oscillator [53]. The dynamics of such two oscilators are fundamentally different. Therefore, the distinct dynamical behaviors of finite duration in the ADM can be used to prove the existence of EPs and distinguish different quantum phases. We employ the Loschmidt echo of quench dynamics to characterize these phase transitions. The Loschmidt echo can be measured experimentally using quantum state tomography [54–56].

The structure of this paper is as follows. In Sec. II, we introduce the model and pointed out the hidden EPs within it. In Sec. III, we solve the Hamiltonian exactly and present the phase diagram of the model. In Sec. IV, we utilize quench dynamics to calculate the Loschmidt echo in order to identify different dynamical phases. Finally, in Sec. V, we provide a summary and discussion. Some details of the calculations are provided in the Appendix.

## II. MODEL AND EXCEPTIONAL POINTS

We consider a Hamiltonian of a single-mode boson coupled to $N$ two-level atoms, where the rotating-wave and counter-rotating-wave terms are distinct. This model is known as the ADM.

$$\begin{aligned}
H &= \omega a^\dagger a + \omega_0 J_z + \frac{g_1}{\sqrt{N}}\left(a^\dagger J_- + a J_+\right) \\
&\quad + \frac{g_2}{\sqrt{N}}\left(a^\dagger J_+ + a J_-\right).
\end{aligned} \tag{1}$$

Here, $a^\dagger$ and $a$ represent the creation and annihilation operators of the single-mode boson, respectively. $J_\pm$ and $J_z$ are the collective atomic operators, and their commutation relations are as follows

$$\left[a, a^\dagger\right] = 1, [J_z, J_\pm] = \pm J_\pm, [J_+, J_-] = 2J_z. \tag{2}$$

The first and second terms of the Hamiltonian represent the free Hamiltonians of the light field and the $N$ two-level atoms, respectively, with their strengths controlled

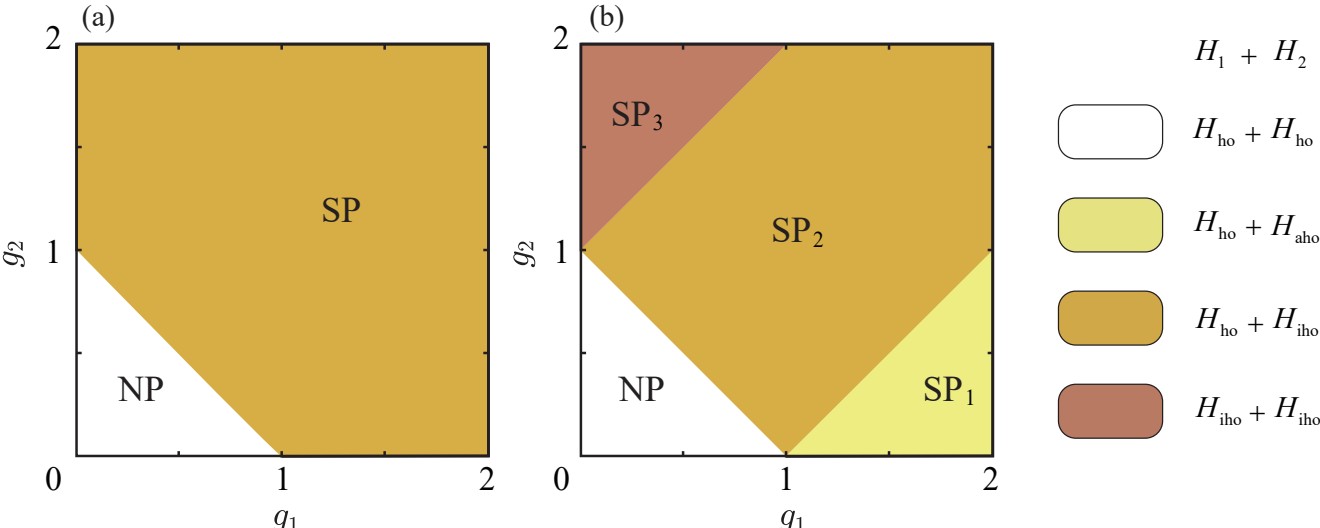

FIG. 1. Phase diagrams of the Hamiltonian in Eq. (1) on the parameter $g_1g_2$ plane, indicating the main conclusion of this work. Different colors in the diagram distinguish different phases of the system. (a) The traditional phase diagram of the anisotropic Dicke model (ADM), obtained by the mean field method, shows that the region $g_1 + g_2 < 1$ corresponds to the normal phase (NP), and the region $g_1 + g_2 > 1$ corresponds to the superradiant phase (SP). (b) The phase diagram of the ADM, revealed by the underlying exceptional points (EPs) of the effective Hamiltonian in Eq. (4) of the system, shows that the original superradiant phase (a) can be further divided into three distinct phases. We label these phases as $SP_1$, $SP_2$, and $SP_3$, respectively. The corresponding equivalent Hamiltonians of the effective Hamiltonian in each region are indicated in the panel. Here, we assume $\omega = \omega_0 = 1$.

by $\omega$ and $\omega_0$. The third and fourth terms correspond to the rotating-wave and counter-rotating-wave coupling terms, with coupling strengths $g_1$ and $g_2$, respectively. When $g_1 = g_2$, the model reduces to the Dicke model. For convenience, in the following derivations, we assume $\omega = \omega_0$, $g_1 > 0$, $g_2 > 0$. The phase diagram of the ADM has been conclusively established in previous studies based on the mean field method [48]. In the parameter plane of $g_1g_2$, the region where $g_1 + g_2 > \omega$ corresponds to the superradiant phase, while the region where $g_1 + g_2 < \omega$ corresponds to the normal phase. The phase diagram is shown in Fig. 1(a).

In the following, we will show that the conventional superradiant phase can be further separated into three regions, in which the systems are characterized by different effective Hamiltonians in large $N$ limit, including the harmonic oscillator, the inverted harmonic oscillator, and their respective counterparts. We refer to these as hierarchic superradiant phases because the same given initial state exhibits distinct dynamic behaviors.

We introduce the Holstein-Primakoff (HP) transformation to convert the spin operators into bosonic operators $b$

$$J_z = b^\dagger b - \frac{N}{2},$$
$$J_+ = (J_-)^\dagger = b^\dagger \sqrt{N - b^\dagger b}, \qquad (3)$$

In the thermodynamic limit where $N \to \infty$ and neglect-

ing constant terms, the Hamiltonian can be rewritten as

$$H_{\text{eff}} = \omega \left(a^\dagger a + b^\dagger b\right) + g_1 \left(a^\dagger b + ab^\dagger\right)$$
$$+ g_2 \left(a^\dagger b^\dagger + ab\right). \qquad (4)$$

$H_{\text{eff}}$ can be regarded as a two-site Hermitian bosonic Kitaev model [45, 46]. In previous studies, we revealed that this model possesses hidden EPs We introduce a linear transformation to decompose $H_{\text{eff}}$ into two independent subspaces Hamiltonian can be written as

$$H_{\text{eff}} = H_1 + H_2$$
$$= \phi_L \begin{pmatrix} h_1 & 0 \\ 0 & h_2 \end{pmatrix} \phi_R, \qquad (5)$$

The non-Hermitian Nambu spinor is defined as $\phi_L = \left(d_1, -d_1^\dagger, d_2, -d_2^\dagger\right)$ and $\phi_R = \left(d_1^\dagger, d_1, d_2^\dagger, d_2\right)^T$. The forms of the two matrices are

$$h_{1,2} = \frac{1}{2}\left(\omega \pm g_1\right)\sigma_z \pm \frac{i}{2}g_2\sigma_y, \qquad (6)$$

$h_{1,2}$ are non-Hermitian matrices, and $\sigma_z$ and $\sigma_y$ are Pauli matrices, defined as

$$\sigma_z = \begin{pmatrix} 1 & 0 \\ 0 & -1 \end{pmatrix}, \sigma_y = \begin{pmatrix} 0 & -i \\ i & 0 \end{pmatrix}. \qquad (7)$$

the eigenvalues of $h_{1,2}$

$$\lambda_1^\pm = \pm\frac{1}{2}\sqrt{(\omega + g_1)^2 - g_2^2},$$
$$\lambda_2^\pm = \pm\frac{1}{2}\sqrt{(\omega - g_1)^2 - g_2^2}. \qquad (8)$$

The corresponding eigenvectors are

$$\phi_1^{\pm} = \begin{pmatrix} -\frac{1}{g_2}\left(\omega + g_1 + 2\lambda_1^{\pm}\right) \\ 1 \end{pmatrix},$$

$$\phi_2^{\pm} = \begin{pmatrix} \frac{1}{g_2}\left(\omega - g_1 + 2\lambda_2^{\pm}\right) \\ 1 \end{pmatrix}. \tag{9}$$

From the forms of the eigenvalues and eigenvectors, we can see that the matrices possess EPs. $h_1$ has a second-order EP when $|\omega + g_1| = |g_2|$, and $h_2$ has a second-order EP when $|\omega - g_1| = |g_2|$. These EP can divide different regions in the $g_1 - g_2$ parameter plane, as shown in Fig. 1(b). In the next section, we will provide the exact solutions for the diagonalized Hamiltonian in each region.

## III. PHASE DIAGRAM

The Hamiltonians $H_1$ and $H_2$ can be explicitly expressed as follows

$$H_1 = (\omega + g_1)d_1^{\dagger}d_1 + \frac{g_2}{2}\left(d_1^{\dagger}d_1^{\dagger} + d_1 d_1\right), \tag{10}$$

and

$$H_2 = (\omega - g_1)d_2^{\dagger}d_2 - \frac{g_2}{2}\left(d_2^{\dagger}d_2^{\dagger} + d_2 d_2\right), \tag{11}$$

respectively. We note that the two Hamiltonians have the same form as

$$\mathcal{H} = \mu\beta^{\dagger}\beta + \frac{\Delta}{2}\left(\beta^{\dagger}\beta^{\dagger} + \beta\beta\right), \tag{12}$$

where $\beta$ is the bosonic annihilation operator. In the Appendix, we provide the derivation of the diagonalization of the Hamiltonian $\mathcal{H}$, based on which two Hamiltonians $H_1$ and $H_2$ can be reduced to different simple form in the four regions in the first quadrant of $g_1 g_2$ plane.

Ignoring the energy constants, there exist three types of equivalent Hamiltonians, given by

$$H_{\mathrm{ho}} = \Omega_i\left(\gamma_i^{\dagger}\gamma_i + \frac{1}{2}\right), \tag{13}$$

$$H_{\mathrm{iho}} = (-1)^{i+1}\frac{\Omega_i}{2}\left[\left(\gamma_i^{\dagger}\right)^2 + \gamma_i^2\right], \tag{14}$$

$$H_{\mathrm{aho}} = -\Omega_i\left(\gamma_i^{\dagger}\gamma_i + \frac{1}{2}\right), \tag{15}$$

with $i = 1, 2$, where $\gamma_i$ are bosonic annihilation operators. The positive factor $\Omega_i$ is givenby

$$\Omega_1 = \sqrt{(\omega + g_1)^2 - g_2^2}, \tag{16}$$

$$\Omega_2 = \sqrt{(\omega - g_1)^2 - g_2^2}. \tag{17}$$

The harmonic oscillator Hamiltonian $H_{\mathrm{ho}}$ is the standard form of the Hamiltonian for a harmonic oscillator. The inverted harmonic oscillator Hamiltonian $H_{\mathrm{iho}}$ represents an inverted harmonic oscillator. This Hamiltonian describes a system with zero potential energy, leading to an unstable system that can tunnel to states with a higher particle number. The anti-harmonic oscillator Hamiltonian $H_{\mathrm{aho}}$ is the negative of the standard harmonic oscillator Hamiltonian. It describes a system where the energy levels are inverted, meaning the ground state has the highest energy and the excited states have lower energies. This is also an stable system. Each of these Hamiltonians has distinct physical properties and implications for the stability and behavior of the system. The harmonic oscillator and anti-harmonic oscillators are stable, while the inverted harmonic oscillators is unstable. In the following, we present the explicit form of the equivalent Hamiltonians in each region.

(i) For $g_1 + g_2 < \omega$, in this region, the two Hamiltonians have the form

$$H_1 = \Omega_1\left(\gamma_1^{\dagger}\gamma_1 + \frac{1}{2}\right) - \frac{1}{2}\left(\omega + g_1\right), \tag{18}$$

and

$$H_2 = \Omega_2\left(\gamma_2^{\dagger}\gamma_2 + \frac{1}{2}\right) - \frac{1}{2}\left(\omega - g_1\right), \tag{19}$$

respectively. Here, $\gamma_1$ and $\gamma_2$ are bosonic annihilation operators, given by

$$\gamma_i = \sinh\theta_i d_i^{\dagger} + \cosh\theta_i d_i, \tag{20}$$

with

$$\tanh\theta_1 = \frac{\omega + g_1 - \Omega_1}{g_2}, \tag{21}$$

and

$$\tanh\theta_2 = \frac{\omega - g_1 - \Omega_2}{g_2}, \tag{22}$$

respectively.

(ii) For $g_1 + g_2 > \omega$ and $g_2 < g_1 - \omega$, in this region, two Hamiltonians have the form

$$H_1 = \Omega_1\left(\gamma_1^{\dagger}\gamma_1 + \frac{1}{2}\right) - \frac{1}{2}\left(\omega + g_1\right), \tag{23}$$

and

$$H_2 = -\Omega_2\left(\gamma_2^{\dagger}\gamma_2 + \frac{1}{2}\right) - \frac{1}{2}\left(\omega - g_1\right), \tag{24}$$

respectively. Here, $\gamma_1$ and $\gamma_2$ have the same forms in Eq. (20), still with

$$\tanh\theta_1 = \frac{\omega + g_1 - \Omega_1}{g_2}, \tag{25}$$

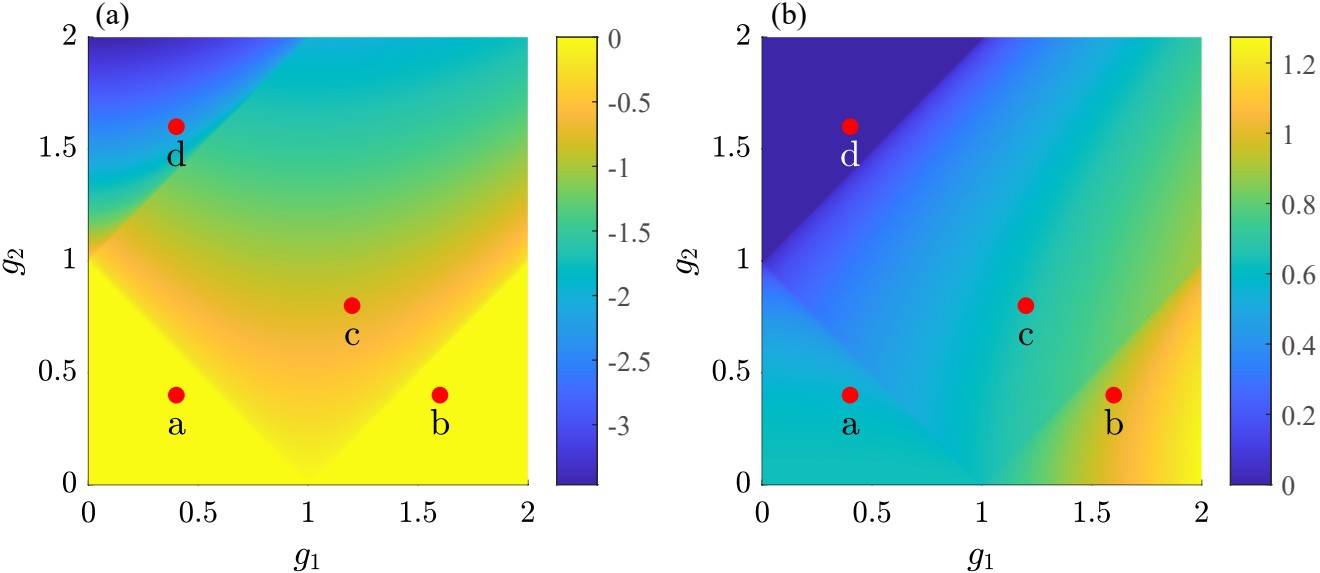

FIG. 2. The plots of the decay rate $\lambda$ in (a), given by Eq. (40) and frequency $f$ in (b), given by Eq. (41) of the effective Hamiltonian on the $g_1 g_2$ plane. It can be seen from the figures that there are clear distinctions between different phases in terms of $\lambda$ and $f$. Four representitive points in each regions are selected, indicated by red dots at the same positions in both panels, with coordinates a$(0.4, 0.4)$, b$(1.6, 0.4)$, c$(1.2, 0.8)$, and d$(0.4, 1.6)$. The corresponding quench dynamical behaviors of the original ADM in finite systems at these points, obtained by numerical simulations, are presented in Fig. 3.

and

$$\tanh \theta_2 = \frac{\omega - g_1 + \Omega_2}{g_2}, \tag{26}$$

respectively.

(iii) For $g_1 + g_2 > \omega$ and $g_1 - \omega < g_2 < g_1 + \omega$, in this region, two Hamiltonians have the form

$$H_1 = \Omega_1 \left( \gamma_1^\dagger \gamma_1 + \frac{1}{2} \right) - \frac{1}{2} (\omega + g_1), \tag{27}$$

and

$$H_2 = i \frac{\Omega_2}{2} \left[ \left( \gamma_2^\dagger \right)^2 + (\gamma_2)^2 \right] - \frac{1}{2} (\omega - g_1), \tag{28}$$

respectively. Here, $\gamma_1$ and $\gamma_2$ have the same forms in Eq. (20), but with

$$\tanh \theta_1 = \frac{(\omega + g_1) - \Omega_1}{g_2}, \tag{29}$$

and

$$\tanh \theta_2 = \frac{g_2 - i\Omega_2}{\omega - g_1}. \tag{30}$$

(iv) For $g_1 + g_2 > \omega$ and $g_1 + \omega < g_2$, in this region, two Hamiltonians have the form

$$H_1 = -i \frac{\Omega_1}{2} \left[ \left( \gamma_2^\dagger \right)^2 + (\gamma_2)^2 \right] - \frac{1}{2} (\omega + g_1), \tag{31}$$

and

$$H_2 = i \frac{\Omega_2}{2} \left[ \left( \gamma_2^\dagger \right)^2 + (\gamma_2)^2 \right] - \frac{1}{2} (\omega - g_1), \tag{32}$$

respectively. Here, $\gamma_1$ and $\gamma_2$ have the same forms in Eq. (20), but with

$$\tanh \theta_1 = \frac{g_2 + i\Omega_1}{\omega + g_1}, \tag{33}$$

and

$$\tanh \theta_2 = \frac{g_2 - i\Omega_2}{\omega - g_1}, \tag{34}$$

respectively. The corresponding equivalent Hamiltonians are indicated in the phase diagram shown in Fig. 1(b). It shows that the configurations of the equivalent Hamiltonians are different in each region. The whole superradiant phase is separated three sub-phases, which are refered to as hierarchic superradiant phases. Here, we would like to emphasize that the phase diagram presented here is not a zero-temperature phase diagram. Different equivalent Hamiltonians exhibit different dynamics, which cannot be captured by mean-field theory. These phases have to be detected by the measurement of information in the excited state. Building upon this insight, we will propose a dynamic demonstration of the phase diagram.

## IV. QUENCH DYNAMICS

In this section, we investigate the dynamic behavior of the phase diagram, including the hierarchical super-

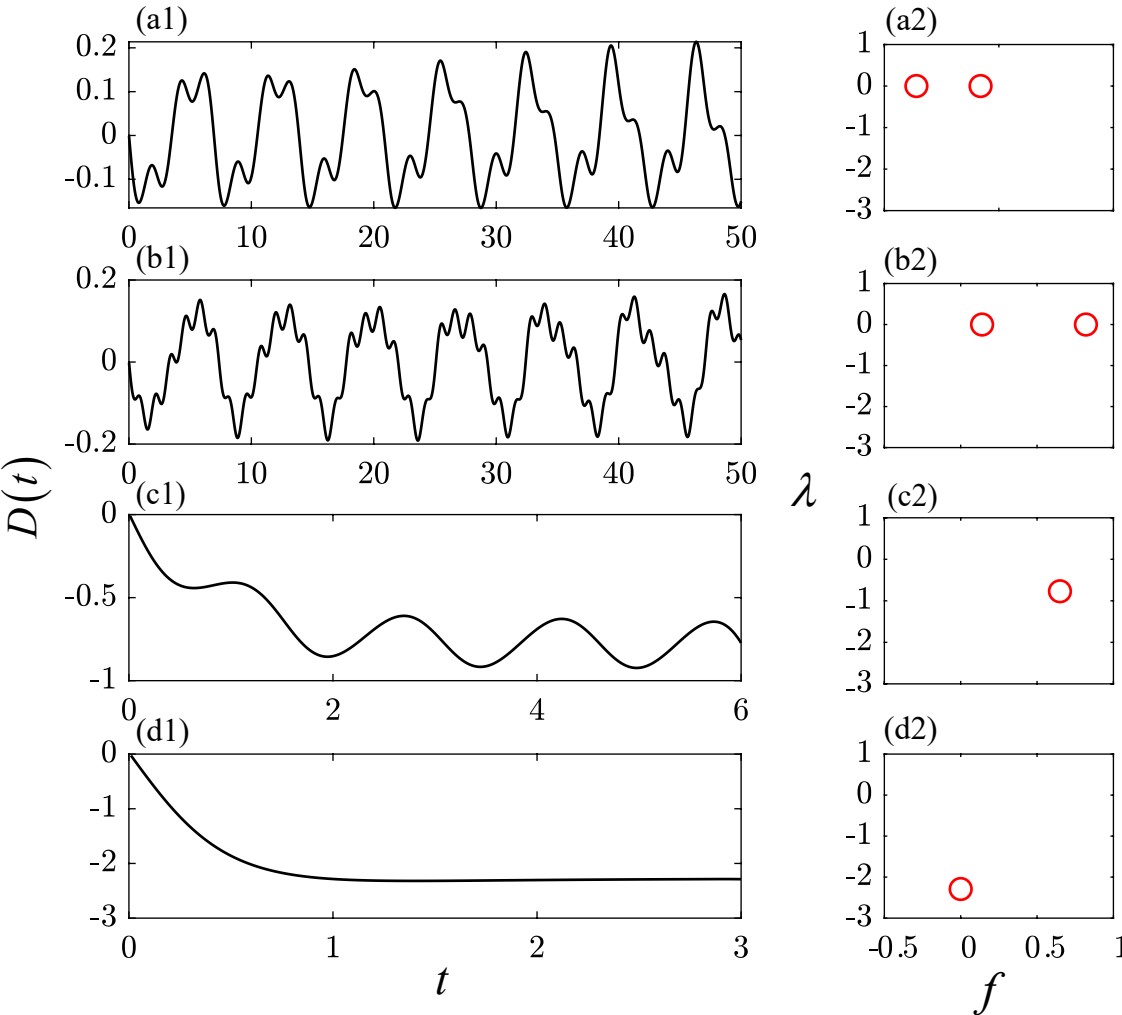

FIG. 3. The plots of $D(t)$, given by Eq. (38), and their characteristics for the original ADM in finite systems at the represented points indicated in Fig. 2. The plots in (a1)-(d1) are obtained by numerical simulations. The corresponding decay rates $\lambda$ and frequencies $f$, plotted in (a2)-(d2), are extracted from the plots of $D(t)$. The number of atoms in the system is $N = 100$. These results are in accordance with the predictions from the analysis of the effective Hamiltonians.

radiant phases. We consider the quench dynamics under the postquench Hamiltonian $H$. We conduct numerical simulations for the Loschmidt echo, defined as

$$L(t) = |\langle \psi(0) | \psi(t) \rangle|^2, \qquad (35)$$

which is a measure of the revival for the initial state $|\psi(0)\rangle$. It allows us to characterize the properties of a system, provided that a proper initial state is chosen. We choose the empty state as the initial state $|\psi(0)\rangle = |\Downarrow\rangle |0\rangle$ and calculate its evolved state

$$|\psi(t)\rangle = \exp(-iHt) |\psi(0)\rangle, \qquad (36)$$

where states $|\Downarrow\rangle$ and $|0\rangle$ are defined by $J_z |\Downarrow\rangle = -N/2 |\Downarrow\rangle$ and $a |0\rangle = 0$, respectively. Before the computation for the finite ADM system, we would like to estimate the possible result.

We start with the investigation for the effective Hamiltonian $H_{\text{eff}}$, which can be dealt with analytically. The corresponding initial state becomes $|\psi(0)\rangle = |0\rangle_a |0\rangle_b$ and evolved state is $|\psi(t)\rangle = \exp(-iH_{\text{eff}}t) |\psi(0)\rangle$, correspondingly. The Loschmidt echo has the following approximate expressions in each regions

$$L(t) = \begin{cases} \left[1 - a\sin^2(\Omega_1 t)\right] \left[1 - b\sin^2(\Omega_2 t)\right], & \text{NP} \\ \left[1 - a\sin^2(\Omega_1 t)\right] \left[1 - b\sin^2(\Omega_2 t)\right], & \text{SP}_1 \\ \left[1 - a\sin^2(\Omega_1 t)\right] \left[\cosh(\Omega_2 t)\right]^{-1}, & \text{SP}_2 \\ \left[\cosh(\Omega_1 t)\right]^{-1} \left[\cosh(\Omega_2 t)\right]^{-1}, & \text{SP}_3 \end{cases} ,$$

$$(37)$$

where the parameters $a$ and $b$ are determined by the values of $g_1$ and $g_2$, satisfying $a, b \in (0, 1)$. In each region of superradiant phases, $L(t)$ is the product of two functions, which take different configurations. For the SP$_1$ region, it is the product of two periodic functions. For the SP$_2$ region, it is the product of a periodic function and a de-

caying function. For the $SP_3$ region, it is the product of two decaying functions. We refer these phases to as hierarchic superradiant phases.

We note that the function $[\cosh(\Omega_i t)]^{-1} \approx 2e^{-\Omega_i t}$, decaying exponentially with rate $\Omega_i$, after long time scale. Then, the oscillating frequency and the decay rate can be the dynamic characters of the hierarchic SPs. In order to characterize the hierarchy of the phases, we focus on the quantity

$$D(t) = \frac{\partial}{\partial t} \ln L(t), \tag{38}$$

because we have

$$\frac{\partial}{\partial t} \ln e^{-\Omega_i t} = -\Omega_i. \tag{39}$$

It is expected that $D(t)$ is the sum of two simple functions, which take different configurations in each region of superradiant phases. Therefore, the factors $\Omega_1$ and $\Omega_2$ can be extracted from the long-time behavior of $D(t)$. For the $SP_1$ region, $D(t)$ oscillates around zero, from which two frequencies $f_1 = \Omega_1/\pi$ and $f_2 = \Omega_2/\pi$ can be extracted. In the $SP_2$ region, it oscillates around a constant, from which the oscillating frequency $f_1$ and the balance point $-\Omega_2$ can be extracted. In the $SP_3$ region, it decays to a constant, from which the decay rate $\lambda = -(\Omega_1 + \Omega_2)$ can be extracted. What is shown in Fig. 2 is the analytical result of the decay rate

$$\lambda = -(\Omega_1 + \Omega_2), \tag{40}$$

and the sum of frequencies

$$f = f_1 + f_2, \tag{41}$$

which can be extracted from the echo of the evolved state of the effective Hamiltonian $H_{\text{eff}}$. We can see the non-analytical behaviors of the plots at the phase boundaries.

Now, we turn to the computation of the corresponding quantities for the original ADM Hamiltonian. For a system with a finite number of atoms, the dimension of the Hilbert space is infinite. Therefore, the time evolution of the initial state is computed using exact diagonalization under the truncation approximation. The computations are performed using a uniform mesh in the time discretization for the trancated matrix. We selected four representative points in the four phases of the ADM to perform quench dynamics verification, and the results are shown in Fig. 3. The extracted decay rate $\lambda$ and frequency $f$ correspond to those in Fig. 2. The results are in accordance with the predictions from the analysis of the effective Hamiltonians. This demonstrates that there indeed exist hierarchical superradiant phases within the traditional superradiant phase of the ADM.

## V. SUMMARY

In summary, we have demonstrated that the conventional superradiant phase can be further separated into three regions. The underlying mechanism is the existence of the exceptional points in the effective Hamiltonians in the thermodynamic limit. Unlike the traditional quantum phase transitions, which usually occur in the ground state of the system, the phase separations arise from the sudden change of the complete set of eigenstates. In this sense, the proposed phase diagram is not merely a mathematical concept, but definitely results in evident observations. Numerical simulations have been performed to compute the Loschmidt echo for finite systems. The results indicate that such observables are sufficient to characterize the hierarchical superradiant phases.

## APPENDIX

In this appendix, we provide the derivation of the diagonalization of the Hamiltonian $\mathcal{H}$, which is equivalent to the two Hamiltonians $H_1$ and $H_2$ given in the main text. The Hamiltonian reads

$$\mathcal{H} = \mu \beta^\dagger \beta + \frac{\Delta}{2}\left(\beta^\dagger \beta^\dagger + \beta\beta\right), \tag{A1}$$

where $\beta$ is the bosonic annihilation operator. Here, we do not restrict the range of $\mu$ and $\Delta$, and $\mathcal{H}$ naturally satisfies

$$\begin{aligned} H_1 &= \mathcal{H}\left(\mu = \omega + g_1, \Delta = g_2\right), \\ H_2 &= \mathcal{H}\left(\mu = \omega - g_1, \Delta = g_2\right). \end{aligned} \tag{A2}$$

We assume that there exists a Bogoliubov transformation

$$\gamma = \sinh\theta\beta^\dagger + \cosh\theta\beta, \tag{A3}$$

that allows for the diagonalization of the Hamiltonian $\mathcal{H}$. Here, $\gamma$ is also the bosonic annihilation operator and the inverse transformation is

$$\beta = \cosh\theta\gamma - \sinh\theta\gamma^\dagger. \tag{A4}$$

The coefficient $\theta$ is determined by the following process. Substituting the transformation into $\mathcal{H}$ we have

$$\begin{aligned} \mathcal{H} &= \frac{1}{2}\left(\Delta\cosh 2\theta - \mu\sinh 2\theta\right)\left[\left(\gamma^\dagger\right)^2 + \gamma^2\right] \\ &\quad + \left(\mu\cosh^2\theta - \frac{\Delta}{2}\sinh 2\theta\right)\gamma^\dagger\gamma \\ &\quad + \left(\mu\sinh^2\theta - \frac{\Delta}{2}\sinh 2\theta\right)\left(1 + \gamma^\dagger\gamma\right). \end{aligned} \tag{A5}$$

We consider the following two cases respectively.

(i) $|\mu| > |\Delta|$, the Hamiltonian can be written as the diagonalized form

$$\mathcal{H} = \text{sgn}(\mu)\left[\sqrt{\mu^2 - \Delta^2}\left(\gamma^\dagger\gamma + \frac{1}{2}\right)\right] - \frac{\mu}{2}, \tag{A6}$$

when we take

$$\tanh\theta = \frac{\mu - \text{sgn}(\mu)\sqrt{\mu^2 - \Delta^2}}{\Delta}. \tag{A7}$$

(ii) $|\mu| < |\Delta|$, the Hamiltonian can be written as the anti-diagonalized form

$$\mathcal{H} = \text{sgn}(\Delta)\frac{1}{2}\{\sqrt{\Delta^2-\mu^2}[(\gamma^\dagger)^2+\gamma^2]\} - \frac{\mu}{2}, \quad \text{(A8)}$$

when we take

$$\tanh\theta = \frac{\Delta - \text{sgn}(\Delta)\sqrt{\Delta^2-\mu^2}}{\mu}. \quad \text{(A9)}$$

**ACKNOWLEDGMENT**

We acknowledge the support of NSFC (Grants No. 12374461).

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
