# Peer review of "Hierarchic superradiant phases in anisotropic Dicke model"

_SciPost Physics Core_

## Round 2 · List of Changes



---

## Round 3 · Referee Report · Anonymous (Referee 2) · 2026-1-7

Report

second reviewer report for "Hierarchic superradiant phases in anisotropic Dicke model"

I am more or less satisfied with the authors response and would recommend publication, provided the authors take the minor comments below into account.

  • Eqns (14-18) and pg. 6-7: What I find still misleading is the term "The positive factors $\Omega_i$ ..." right before (17) as for large |g_2| > |\omega+-g_1|, the "positive factors" become imaginary, in particular for the limits discussed in Eqns (27,30,31). Does it become correct when you simply write "The factors $\Omega_i$ are given by"?

  • Figures: Optionally, the authors may consider to plot vs. dimensionless parameters, e.g. g_1/\omega, g_2/\omega in Figs. 1/2 and vs. \omega t in Fig. 3

Recommendation

Ask for minor revision

  • validity: -
  • significance: -
  • originality: -
  • clarity: -
  • formatting: -
  • grammar: -

Author:  DaKai He  on 2026-01-08  [id 6209]

(in reply to Report 1 on 2026-01-07)
Category:
answer to question

List of changes:
(1) The forms of Eqs. (17) and (18) have been revised.
(2) Figures 1, 2, and 3 have been represented in dimensionless parameters.

We thank the Referee for the time and effort dedicated to evaluating our manuscript. Our responses to the comments are as follows:

(1) We appreciate the Referee's careful reading and valuable suggestion. The purpose of presenting Hamiltonians (14)–(16) is to illustrate the differences among the three types. Therefore, it is necessary to require that the factors Ωi remain positive. In the revised version, we have added absolute values to expressions (17) and (18).

(2) We are very grateful for this practical suggestion. In the updated version, we have adjusted the axes to be represented in dimensionless parameters.

Attachment:

ADM_v4.pdf

---

## Round 3 · List of Changes

List of changes: 1. More references have been added. 2. A discussion on negative energy has been added. 3. The discussion on stability has been rephrased. 4. The discussions on the truncation of the Hilbert space have been added. 5. A discussion on the impact of the initial state on the phase boundaries has been added. The main revised text is marked in red in the manuscript.

---

## Editorial Decision

voting_in_preparation